# ON THE NEURAL TANGENT KERNEL OF EQUILIBRIUM MODELS

## ABSTRACT

Existing analyses of the neural tangent kernel (NTK) for infinite-depth networks show that the kernel typically becomes degenerate as the number of layers grows. This raises the question of how to apply such methods to practical "infinite depth" architectures such as the recently-proposed deep equilibrium (DEQ) model, which directly computes the infinite-depth limit of a weight-tied network via root-finding. In this work, we show that because of the input injection component of these networks, DEQ models have non-degenerate NTKs even in the infinite depth limit. Furthermore, we show that these kernels themselves can be computed by an analogous root-finding problem as in traditional DEQs, and highlight methods for computing the NTK for both fully-connected and convolutional variants. We evaluate these models empirically, showing they match or improve upon the performance of existing regularized NTK methods.

## 1 INTRODUCTION

Recent works empirically observe that as the depth of a weight-tied input-injected network increases, its output tends to converge to a fixed point. Motivated by this phenomenon, DEQ models were proposed to effectively represent an "infinite depth" network by root-finding. A natural question to ask is, what will DEQs become if their widths also go to infinity? It is well-known that at certain random initialization, neural networks of various structures converge to Gaussian processes as their widths go to infinity (Neal, 1996; Lee et al., 2017; Yang, 2019; Matthews et al., 2018; Novak et al., 2018; Garriga-Alonso et al., 2018). Recent deep learning theory advances have also shown that in the infinite width limit, with proper initialization (the NTK initialization), training the network $f_\theta$ with gradient descent is equivalent to solving kernel regression with respect to the neural tangent kernel (Arora et al., 2019; Jacot et al., 2018; Yang, 2019; Huang et al., 2020).

However, as the depth goes to infinity, Jacot et al. (2019) showed that the NTKs of fully-connected neural networks (FCNN) converge either to a constant (*freeze*), or to the Kronecker Delta (*chaos*). In this work, we show that with input injection, the DEQ-NTKs converge to meaningful fixed points that depend on the input in a non-trivial way, thus avoiding both freeze and chaos. Furthermore, analogous to DEQ models, we can compute these kernels by solving an analogous fixed point equation, rather than simply iteratively applying the updates associated with the traditional NTK. Moreover, such derivations carry over to other structures like convolution DEQs (CDEQ) as well. We evaluate the approach and demonstrate that it typically matches or improves upon the performance of existing regularized NTK methods.

## 2 BACKGROUND AND PRELIMINARIES

Bai et al. (2019) proposed the DEQ model, which is equivalent to running an infinite depth network with tied weight and input injection. These methods trace back to some of the original work in recurrent backpropagation (Almeida, 1990; Pineda, 1988), but with specific emphasis on: 1) computing the fixed point directly via root-finding rather than forward iteration; and 2) incorporating the elements from modern deep networks in the single "layer", such as self-attention transformers (Bai et al., 2019), multi-scale convolutions (Bai et al., 2020), etc. The DEQ algorithm finds the infinite depth fixed point using quasi-Newton root finding methods, and then backpropagates using implicit differentiation without storing the derivatives in the intermediate layers, thus achieving a constant

memory complexity. Furthermore, although a traditional DEQ model does not always guarantee to find a stable fixed point, with careful parameterization and update method, monotone operator DEQs can ensure the existence of a unique stable fixed point (Winston & Kolter, 2020).

On the side of connecting neural networks to kernel methods, Neal (1996) first discovered that a single-layered network with randomly initialized parameters becomes a Gaussian process (GP) in the large width limit. Such connection between neural networks and GP was later extended to multiple layers (Lee et al., 2017; Matthews et al., 2018) and various other architectures (Yang, 2019; Novak et al., 2018; Garriga-Alonso et al., 2018). The networks studied in this line of works are randomly initialized, and one can imagine these networks as having fixed parameters throughout the training process, except for the last classification layer. Following the naming convention of Arora et al. (2019), we call these networks *weakly-trained*, and networks where every layer is updated are called *fully-trained*. Weakly-trained nets induce the kernel $\Theta(x, y) = \mathbb{E}_{\theta \sim \mathcal{N}}[f(\theta, x) \cdot f(\theta, y)]$, where $x, y \in \mathbb{R}^d$ are two samples, $\theta$ represents the parameters of the network, $\mathcal{N}$ is the initialization distribution (often Gaussian) over $\theta$, and $f(\theta, \cdot) \in \mathbb{R}$ is the output of the network.

One related topic in studying the relation between Gaussian process kernel and depth is the mean-field theory. Poole et al. (2016); Schoenholz et al. (2016) showed that the correlations between all inputs on an infinitely wide weakly-trained net become either perfectly correlated (*order*) or decorrelated (*chaos*) as depth increases. This aligns with the observation in Jacot et al. (2019). They suggested we should initialize the neural network on the "edge-of-chaos" to make sure that signals can propagate deep enough in the forward direction, and the gradient does not vanish or explode during backpropagation (Raghu et al., 2017; Schoenholz et al., 2016). These mean-field behaviors were later proven for various other structures like RNNs, CNNs, and NTKs as well (Chen et al., 2018a; Xiao et al., 2018; Gilboa et al., 2019; Hayou et al., 2019). We emphasize that despite the similar appearance, our setting avoids the order vs. chaos scheme completely by adding input injection. Such structure guarantees the converged nets depend nontrivially on the inputs, as we will see later in the experiments.

It can be unsatisfying that the previous results only involve weakly-trained nets. Interestingly, similar limiting behavior was proven by Jacot et al. (2018) to hold for fully-trained networks as well. They showed the kernel induced by a fully-trained infinite width network is the following:

$$\Theta(x, y) = \mathbb{E}_{\theta \sim \mathcal{N}} \left[ \left\langle \frac{\partial f(\theta, x)}{\partial \theta}, \frac{\partial f(\theta, y)}{\partial \theta} \right\rangle \right]. \tag{1}$$

They also gave a recursive formulation for the NTK of FCNN. Arora et al. (2019); Yang (2020) later provided formulation for convolution NTK and other structures.

One may ask what happens if both the width and the depth go to infinity. It turns out that the vanilla FCNN does not have a meaningful convergence: either it gives constant kernels or Kronecker Delta kernels (Jacot et al., 2019). On the bright side, this assertion is not always the case for other network structures. For example, the NTK induced by ResNet provides a meaningful fixed point in the large depth limit (Huang et al., 2020). This may seem to give one explanation why ResNet outperforms FCNN, but unfortunately they also show that the ResNet NTK with infinite depth is no different from the ResNet NTK with just depth one. This conclusion makes the significance of infinite depth questionable.

## 2.1 NOTATIONS

Throughout the paper, we write $\theta$ as the parameters for some network $f_\theta$ or equivalently, $f(\theta, \cdot)$. We write capital letter $W$ to represent matrices or tensors, which should be clear from the context, and use $[W]_i$ to represent the element of $W$ indexed by $i$. We write lower case letter $x$ to represent vectors or scalars. For $a \in \mathbb{Z}_+$, let $[a] = \{1, \ldots, a\}$. Denote $\sigma(x) = \sqrt{2} \max(0, x)$ as the normalized ReLU and $\dot{\sigma}$ its derivative (which only need to be well-defined almost everywhere). The symbol $\sigma_a^2$ with subscript is always used to denote the variance of some distribution. We write $\mathcal{N}(\mu, \Sigma)$ as the Gaussian distribution with mean $\mu \in \mathbb{R}^d$ and covariance matrix $\Sigma \in \mathbb{R}^{d \times d}$. We let $\mathbb{S}^{d-1}$ be the unit sphere embedded in $\mathbb{R}^d$.

## 3 DEQ-NTK OF FULLY-CONNECTED NEURAL NETWORKS

In this section, we show how to derive the NTK of the fully-connected DEQ (DEQ-NTK). From now on, we simplify fully-connected DEQs as DEQs. Recall that DEQs are equivalent to infinitely deep fully-connected neural nets with input injection (FCNN-IJ), and one can either exactly solve the fixed point using root-finding (up to machine epsilon and root-finding algorithm accuracy) or approximate the DEQs by just doing finite depth forward iterations. In section 3.1, we show the NTK of the approximated DEQ using finite depth iteration, and in section 3.2 we demonstrate how to get the exact convergence point of DEQ-NTK. The details of this section can be found in appendix A.

### 3.1 FINITE DEPTH ITERATION OF DEQ-NTK

Let $d$ be the input dimension, $x, y \in \mathbb{R}^d$ be a pair of inputs, $N_h$ be the width of the $h^{th}$ layers where $h \in [L+1]$. Let $N_0 = d$ and $N_{L+1} = 1$. Define the FCNN-IJ with $L$ hidden layers as follows:

$$f_\theta^{(h)}(x) = \sqrt{\frac{\sigma_W^2}{N_h}} W^{(h)} g^{(h-1)}(x) + \sqrt{\frac{\sigma_U^2}{N_h}} U^{(h)} x + \sqrt{\frac{\sigma_b^2}{N_h}} b^{(h)}, \ h \in [L+1]$$

$$g^{(L)}(x) = \sigma(f_\theta^{(L)}(x))$$

where $W^{(h)} \in \mathbb{R}^{N_h \times N_{h-1}}$, $U^{(h)} \in \mathbb{R}^{N_h \times d}$ are the internal weights, and $b^{(h)} \in \mathbb{R}^{N_h}$ are the bias terms. These parameters are chosen using the NTK initialization. Let us pick $\sigma_W, \sigma_U, \sigma_b \in \mathbb{R}$ arbitrarily in this section.

**NTK initialization.** We randomly initialize every entry of every $W, U, b$ from $\mathcal{N}(0, 1)$.

Without loss of generality (WLOG) we assume the width of the hidden layer $N_h = N$ is the same across different layers. We remark the readers to distinguish FCNN-IJ from a recurrent neural network (RNN): our model injects the original input to each layer, whereas a RNN has a sequence of input $(x_1, \dots, x_T)$, and inject $x_t$ to the $t^{th}$-layer.

Here is a crucial distinction between finite width DEQs and infinite width DEQs:

**Remark 1.** In the finite width regime, one typically has to assume the DEQs have tied weights, that is, $W^{(1)} = \dots = W^{(L+1)}$. Otherwise it is unlikely the network will converge at all. In fact, one needs to be very careful with the parametrization of the weights to guarantee the fixed point is unique and stable. This is not the case in the infinite width regime. As we shall see soon, even with distinct weights in each layer, the convergence of DEQ-NTKs only depend on $\sigma_W^2, \sigma_U^2, \sigma_b^2$, and the nonlinearity $\sigma$. Assuming untied weights makes the analysis easier, but the same argument can be made rigorously for tied weights as well, see Yang (2019; 2020).

Our main theorem is the following:

**Theorem 1.** *Recursively define the following quantities for $h \in [L]$:*

$$\Sigma^{(0)}(x, y) = x^\top y \tag{2}$$

$$\Lambda^{(h)}(x, y) = \begin{pmatrix} \Sigma^{(h-1)}(x, x) & \Sigma^{(h-1)}(x, y) \\ \Sigma^{(h-1)}(y, x) & \Sigma^{(h-1)}(y, y) \end{pmatrix} \in \mathbb{R}^{2 \times 2} \tag{3}$$

$$\Sigma^{(h)}(x, y) = \sigma_W^2 \mathbb{E}_{(u,v) \sim \mathcal{N}(0, \Lambda^{(h)})}[\sigma(u)\sigma(v)] + \sigma_U^2 x^\top y + \sigma_b^2 \tag{4}$$

$$\dot{\Sigma}^{(h)}(x, y) = \sigma_W^2 \mathbb{E}_{(u,v) \sim \mathcal{N}(0, \Lambda^{(h)})}[\dot{\sigma}(u)\dot{\sigma}(v)] \tag{5}$$

*Then the L-depth iteration to the DEQ-NTK can be expressed as:*

$$\Theta^{(L)}(x, y) = \sum_{h=1}^{L+1} \left( \left( \Sigma^{(h-1)}(x, y) \right) \cdot \prod_{h'=h}^{L+1} \dot{\Sigma}^{(h')}(x, y) \right), \tag{6}$$

*where by convention we set $\dot{\Sigma}^{L+1}(x, y) = 1$ for the L-depth iteration.*

*Proof Sketch.* The first step is to show that at each layer $h \in [L]$, the representation $f_\theta^{(h)}(x)$ is associated with a Gaussian process with kernel eq. (3) as $N \to \infty$. Then use the characterization in eq. (1), calculate the NTK by:

$$\Theta^{(L)}(x, y) = \mathbb{E}_\theta \left[ \left\langle \frac{\partial f(\theta, x)}{\partial \theta}, \frac{\partial f(\theta, y)}{\partial \theta} \right\rangle \right]$$

$$= \underbrace{\mathbb{E}_\theta \left[ \left\langle \frac{\partial f(\theta, x)}{\partial W}, \frac{\partial f(\theta, y)}{\partial W} \right\rangle \right]}_{①} + \underbrace{\mathbb{E}_\theta \left[ \left\langle \frac{\partial f(\theta, x)}{\partial U}, \frac{\partial f(\theta, y)}{\partial U} \right\rangle \right]}_{②} + \underbrace{\mathbb{E}_\theta \left[ \left\langle \frac{\partial f(\theta, x)}{\partial b}, \frac{\partial f(\theta, y)}{\partial b} \right\rangle \right]}_{③}.$$

Calculating each term using the chain rule, we get eq. (6). $\qquad\square$

## 3.2 FIXED POINT OF DEQ-NTK

Based on eq. (6), we are now ready to answer what the fixed point of $\Theta^{(L)}$ is. By convention, we assume the two samples $x, y \in \mathbb{S}^{d-1}$, and we require the parameters $\sigma_W^2, \sigma_U^2, \sigma_b^2$ obey the DEQ-NTK initialization.

**DEQ-NTK initialization.** Let every entry of every $W, U, b$ follows the NTK initialization described in section 3.1, as well as the additional requirement $\sigma_W^2 + \sigma_U^2 + \sigma_b^2 = 1$.

Let the nonlinear activation function $\sigma$ be the normalized ReLU: $\sigma(x) = \sqrt{2} \max(0, x)$.

**Definition 3.1** (Normalized activation). We call an activation function $\sigma : \mathbb{R} \to \mathbb{R}$ normalized if $\mathbb{E}_{x \sim \mathcal{N}(0,1)}[\sigma(x)^2] = 1$.

Using normalized activations along with DEQ-NTK initialization, we can derive the main convergence theorem:

**Theorem 2.** *Use same notations and settings in theorem 1. With input data $x, y \in \mathbb{S}^{d-1}$, parameters $\sigma_W^2, \sigma_U^2, \sigma_b^2$ follow the DEQ-NTK initialization, the fixed point of DEQ-NTK is*

$$\Theta^*(x, y) \triangleq \lim_{L \to \infty} \Theta^{(L)}(x, y) = \frac{\Sigma^*(x, y)}{1 - \dot{\Sigma}^*(x, y)}, \tag{7}$$

*where $\Sigma^*(x, y) \triangleq \rho^*$ is the root of:*

$$R_\sigma(\rho) - \rho, \text{ where } R_\sigma(\rho) \triangleq \sigma_W^2 \left( \frac{\sqrt{1 - \rho^2} + (\pi - \cos^{-1} \rho) \rho}{\pi} \right) + \sigma_U^2 x^\top y + \sigma_b^2, \tag{8}$$

*and*

$$\dot{\Sigma}^*(x, y) \triangleq \lim_{h \to \infty} \dot{\Sigma}^{(h)}(x, y) = \sigma_W^2 \left( \frac{\pi - \cos^{-1}(\rho^*)}{\pi} \right). \tag{9}$$

*Proof.* Due to the fact that $x \in \mathbb{S}^{d-1}$, $\sigma$ being a normalized activation, and DEQ-NTK initialization, one can easily calculate by induction that for all $h \in [L]$:

$$\Sigma^{(h)}(x, x) = \sigma_W^2 \mathbb{E}_{u \sim \mathcal{N}(0,1)}[\sigma(u)^2] + \sigma_V^2 x^\top x + \sigma_b^2 = 1$$

This indicates that in eq. (3), the covariance matrix has a special structure $\Lambda^{(h)}(x, y) = \begin{pmatrix} 1 & \rho \\ \rho & 1 \end{pmatrix}$, where $\rho = \Sigma^{(h-1)}(x, y)$ depends on $h, x, y$. For simplicity we omit the $h, x, y$ in $\Lambda^{(h)}(x, y)$. As shown in Daniely et al. (2016):

$$\mathbb{E}_{(u,v) \sim \mathcal{N}(0,\Lambda)}[\sigma(u)\sigma(v)] = \frac{\sqrt{1 - \rho^2} + (\pi - \cos^{-1}(\rho)) \rho}{\pi} \tag{10}$$

$$\mathbb{E}_{(u,v) \sim \mathcal{N}(0,\Lambda)}[\dot{\sigma}(u)\dot{\sigma}(v)] = \frac{\pi - \cos^{-1}(\rho)}{\pi} \tag{11}$$

Adding input injection and bias, we derive eq. (8) from eq. (10), and similarly, eq. (9) from eq. (11). Notice that iterating eqs. (2) to (4) to solve for $\Sigma^{(h)}(x, y)$ is equivalent to iterating $(R_\sigma \circ \cdots \circ R_\sigma)(\rho)$ with initial input $\rho = x^\top y$. Take the derivative

$$\left| \frac{dR_\sigma(\rho)}{d\rho} \right| = \left| \sigma_W^2 \left( 1 - \frac{\cos^{-1}(\rho)}{\pi} \right) \right| < 1, \text{ if } \sigma_W^2 < 1 \text{ and } -1 \le \rho < 1.$$

For $x \ne y$ we have $-1 \le \rho < c < 1$ for some $c$ (this is because we only have finite number of inputs $x, y$) and by DEQ-NTK initialization we have $\sigma_W^2 < 1$, so the above inequality hold. Hence $R_\sigma(\rho)$ is a contraction on $[0, c]$, and we conclude that the fixed point $\rho^*$ is attractive.

By lemma 1, if $\sigma_W^2 < 1$, then the limit of eq. (6) exists, so we can rewrite the summation form in eq. (6) in a recursive form:

$$\Theta^{(0)}(x, y) = \Sigma^{(0)}(x, y)$$
$$\Theta^{(L+1)}(x, y) = \dot{\Sigma}^{(L+1)}(x, y) \cdot \Theta^{(L)}(x, y) + \Sigma^{(L+1)}(x, y),$$

and directly solve the fixed point iteration:

$$\lim_{L \to \infty} \Theta^{(L+1)}(x, y) = \lim_{L \to \infty} \left( \dot{\Sigma}^{(L+1)}(x, y) \cdot \Theta^{(L)}(x, y) + \Sigma^{(L+1)}(x, y) \right)$$
$$\implies \lim_{L \to \infty} \Theta^{(L+1)}(x, y) = \dot{\Sigma}^*(x, y) \cdot \lim_{L \to \infty} \Theta^{(L)}(x, y) + \Sigma^*(x, y) \quad (12)$$
$$\implies \lim_{L \to \infty} \Theta^{(L)}(x, y) = \dot{\Sigma}^*(x, y) \cdot \lim_{L \to \infty} \Theta^{(L)}(x, y) + \Sigma^*(x, y).$$

Solving for $\lim_{L \to \infty} \Theta^{(L)}(x, y)$ we get $\Theta^*(x, y) = \frac{\Sigma^*(x,y)}{1 - \dot{\Sigma}^*(x,y)}$.

$\square$

**Remark 2.** Note our $\Sigma^*(x, y)$ always depends on the inputs $x$ and $y$, so the information between two inputs is always preserved, even if the depth goes to infinity. On the contrary as pointed out by Jacot et al. (2019), without input injection, $\Sigma^{(h)}(x, y)$ always converges to 1 as $h \to \infty$, even if $x \ne y$.

## 4 DEQ WITH CONVOLUTION LAYERS

In this section we show how to derive the NTKs for convolution DEQs (CDEQ). Although in this paper only the CDEQ with vanilla convolution structure is considered in experiments, we remark that our derivation is general enough for other CDEQ structure as well, for instance, CDEQ with global pooling layer. The details of this section can be found in appendix B.

Unlike the FCNN-IJ, whose intermediate NTK representation is a real number. For convolution neural networks (CNN), the intermediate NTK representation is a four-way tensor. In the following, we will present the notations, CNN with input injection (CNN-IJ) formulation, the CDEQ-NTK initialization, and our main theorem.

**Notation.** We adopt the notations from Arora et al. (2019). Let $x, y \in \mathbb{R}^{P \times Q}$ be a pair of inputs, let $q \in \mathbb{Z}_+$ be the filter size (WLOG assume it is odd as well). By convention, we always pad the representation (both the input layer and hidden layer) with 0's. Denote the convolution operation as following:

$$[w * x]_{ij} = \sum_{a=-\frac{q-1}{2}}^{\frac{q-1}{2}} \sum_{b=-\frac{q-1}{2}}^{\frac{q-1}{2}} [w]_{a+\frac{q+1}{2}, b+\frac{q+1}{2}} [x]_{a+i, b+j} \text{ for } i \in [P], j \in [Q].$$

Denote

$$\mathcal{D}_{ij,i'j'} = \Big\{ (i+a, j+b, i'+a', j'+b') \in [P] \times [Q] \times [P] \times [Q] :$$
$$- (q-1)/2 \le a, b, a', b' \le (q-1)/2 \Big\}.$$

Intuitively, $\mathcal{D}_{ij,i'j'}$ is a $q \times q \times q \times q$ set of indices centered at $(ij, i'j')$. For any tensor $T \in \mathbb{R}^{P \times Q \times P \times Q}$, let $[T]_{\mathcal{D}_{ij,i'j'}}$ be the natural sub-tensor and let $\text{Tr}(T) = \sum_{i,j} T_{ij,ij}$.

**Formulation of CNN-IJ.**  Define the CNN-IJ as follows:

- Let the input $x^{(0)} = x \in \mathbb{R}^{P \times Q \times C_0}$, where $C_0$ is the number of input channels, and $C_h$ is the number of channels in layer $h$. Assume WLOG that $C_h = C$ for all $h \in [L]$

- For $h = 1, \ldots, L$, let the inner representation

$$\tilde{x}^{(h)}_{(\beta)} = \sum_{\alpha=1}^{C_{h-1}} \sqrt{\frac{\sigma_W^2}{C_h}} W^{(h)}_{(\alpha),(\beta)} * x^{(h-1)}_{(\alpha)} + \sum_{\alpha=1}^{C_0} \sqrt{\frac{\sigma_U^2}{C_h}} U^{(h)}_{(\alpha),(\beta)} * x^{(0)}_{(\alpha)} \qquad (13)$$

$$\left[ x^{(h)}_{(\beta)} \right]_{ij} = \frac{1}{[S]_{ij}} \left[ \sigma \left( \tilde{x}^{(h)}_{(\beta)} \right) \right]_{ij}, \text{ for } i \in [P], j \in [Q] \qquad (14)$$

where $W^{(h)}_{(\alpha),(\beta)} \in \mathbb{R}^{q \times q}$ represent the convolution operator from the $\alpha^{th}$ channel in layer $h - 1$ to the $\beta^{th}$ channel in layer $h$. Similarly, $U^{(h)}_{(\alpha),(\beta)} \in \mathbb{R}^{q \times q}$ injects the input in each convolution window. $S \in \mathbb{R}^{P \times Q}$ is a normalization matrix. Let $W, U, S, \sigma_U^2, \sigma_W^2$ be chosen by the CDEQ-NTK initialization described later. Notice here we assume WLOG that the number of channels in the hidden layers is the same.

- The final output is defined to be $f_\theta(x) = \sum_{\alpha=1}^{C_L} \left\langle W^{(L+1)}_{(\alpha)}, x^{(L)}_{(\alpha)} \right\rangle$, where $W^{(L+1)}_{(\alpha)} \in \mathbb{R}^{P \times Q}$ is sampled from standard Gaussian distribution.

**CDEQ-NTK initialization.**  Let $1_q \in \mathbb{R}^{q \times q}, X \in \mathbb{R}^{P \times Q}$ be two all-one matrices. Let $\tilde{X} \in \mathbb{R}^{(P+2) \times (Q+2)}$ be the output of zero-padding $X$. We index the rows of $\tilde{X}$ by $\{0, 1, \ldots, P+1\}$ and columns by $\{0, 1, \ldots, Q+1\}$. For position $i \in [P], j \in [Q]$, let $\left([S]_{ij}\right)^2 = [1_q * \tilde{X}]_{ij}$ in eq. (14). Let every entry of every $W, U$ be sampled from $\mathcal{N}(0, 1)$ and $\sigma_W^2 + \sigma_U^2 = 1$.

Using the above defined notations, we now state the CDEQ-NTK.

**Theorem 3.** *Let $x, y \in \mathbb{R}^{P \times Q \times C_0}$ be s.t $\|x_{ij}\|_2 = \|y_{ij}\|_2 = 1$ for $i \in [P], j \in [Q]$. Define the following expressions recursively (some $x, y$ are omitted in the notations), for $(i, j, i', j') \in [P] \times [Q] \times [P] \times [Q], h \in [L]$*

$$K^{(0)}_{ij,i'j'}(x, y) = \left[ \sum_{\alpha \in [C_0]} x_{(\alpha)} \otimes y_{(\alpha)} \right]_{ij,i'j'} \qquad (15)$$

$$\left[ \Sigma^{(0)}(x, y) \right]_{ij,i'j'} = \frac{1}{[S]_{ij}[S]_{i'j'}} \sum_{\alpha=1}^{C_0} \text{Tr} \left( \left[ K^{(0)}_{(\alpha)}(x, y) \right]_{\mathcal{D}_{ij,i'j'}} \right) \qquad (16)$$

$$\Lambda^{(h)}_{ij,i'j'}(x, y) = \begin{pmatrix} \left[ \Sigma^{(h-1)}(x, x) \right]_{ij,ij} & \left[ \Sigma^{(h-1)}(x, y) \right]_{ij,i'j'} \\ \left[ \Sigma^{(h-1)}(y, x) \right]_{i'j',ij} & \left[ \Sigma^{(h-1)}(y, y) \right]_{i'j',i'j'} \end{pmatrix} \in \mathbb{R}^{2 \times 2} \qquad (17)$$

$$\left[ K^{(h)}(x, y) \right]_{ij,i'j'} = \frac{\sigma_W^2}{[S]_{ij} \cdot [S]_{i'j'}} \mathop{\mathbb{E}}_{(u,v) \sim \mathcal{N}(0, \Lambda^{(h)}_{ij,i'j'})} [\sigma(u)\sigma(v)] + \frac{\sigma_U^2}{[S]_{ij} \cdot [S]_{i'j'}} [K^{(0)}]_{ij,i'j'} \qquad (18)$$

$$\left[ \dot{K}^{(h)}(x, y) \right]_{ij,i'j'} = \frac{\sigma_W^2}{[S]_{ij} \cdot [S]_{i'j'}} \mathop{\mathbb{E}}_{(u,v) \sim \mathcal{N}(0, \Lambda^{(h)}_{ij,i'j'})} [\dot{\sigma}(u)\dot{\sigma}(v)] \qquad (19)$$

$$\left[ \Sigma^{(h)}(x, y) \right]_{ij,i'j'} = \text{Tr} \left( \left[ K^{(h)}(x, y) \right]_{\mathcal{D}_{ij,i'j'}} \right) \qquad (20)$$

*Define the linear operator $\mathcal{L} : \mathbb{R}^{P \times Q \times P \times Q} \to \mathbb{R}^{P \times Q \times P \times Q}$ via $[\mathcal{L}(M)]_{ij,i'j'} = \text{Tr} \left( [M]_{\mathcal{D}_{ij,i'j'}} \right)$.*

*Then the CDEQ-NTK can be found solving the following linear system:*

$$\Theta^*(x, y) = \dot{K}^*(x, y) \odot \mathcal{L}(\Theta^*(x, y)) + K^*(x, y), \qquad (21)$$

*where $K^*(x, y) = \lim_{L \to \infty} K^{(L)}(x, y), \dot{K}^*(x, y) = \lim_{L \to \infty} \dot{K}^{(L)}(x, y)$. The limit exists if $\sigma_W^2 < 1$.*

## 5 EXPERIMENTS

In this section, we evaluate the performance of DEQ-NTK and CDEQ-NTK on both MNIST and CIFAR-10 datasets. We also compare the performance of finite depth NTK and finite depth iteration of DEQ-NTK.

**Implementation.** For DEQ-NTK, aligned with the theory, we normalize the dataset such that each data point has unit length. The fixed point $\Sigma^*(x, y)$ is solved by using the modified Powell hybrid method (Powell, 1970). Notice these root finding problems are one-dimensional, hence can be quickily solved. For CDEQ-NTK, the input data $x$ has dimension $P \times Q \times C_0$, and we normalize $x$ s.t $\|x_{ij}\|_2 = 1$ for any $i \in [P], j \in [Q]$. We set $q = 3$ and stride 1. The fixed point $\Sigma^*(x, y) \in \mathbb{R}^{P \times Q \times P \times Q}$ is approximated by running 20 iterations of eq. (17), eq. (18), and eq. (20). The actual CDEQ-NTK $\Theta(x, y)$ is then calculated by solving the sparse linear system eq. (21).

After obtaining the NTK matrix, we apply kernel regressions (without regularization unless stated otherwise). For any label $y \in \{1, \ldots, n\}$, denote its one-hot encoding by $\mathbf{e}_y$. Let $\mathbf{1} \in \mathbb{R}^n$ be an all-1 vector, we train on the new encoding $-0.1 \cdot \mathbf{1} + \mathbf{e}_y$. That is, we change the "1" to 0.9, and the "0" to $-0.1$, as suggested by Novak et al. (2018).

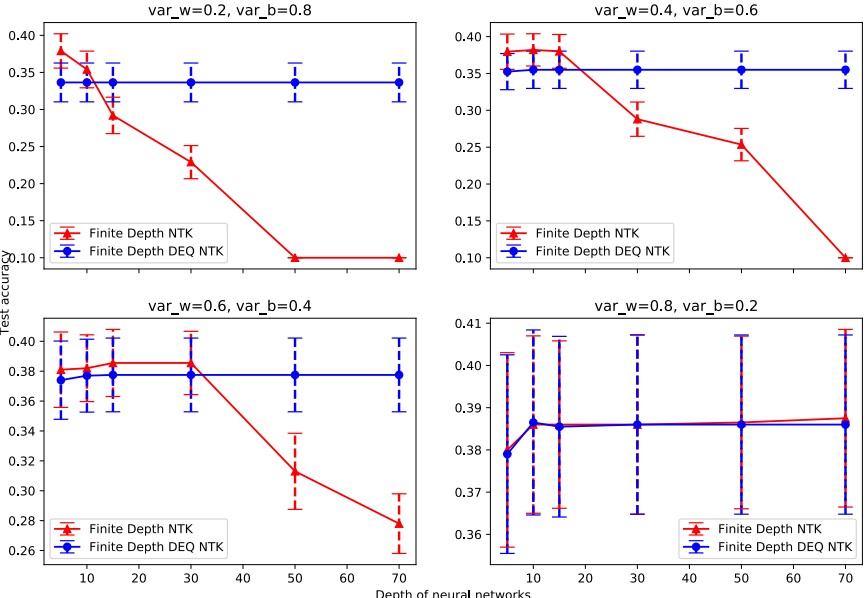

Figure 1: Finite depth NTK vs. finite depth iteration of DEQ-NTK. In all experiments, the NTK is initialized with $\sigma_W^2$ and $\sigma_b^2$ in the title. For DEQ-NTK we set $\sigma_U^2 = \sigma_b^2 - 0.1$ in the title, and the actual $\sigma_b^2 = 0.1$. All models are trained on 1000 CIFAR-10 data and tested on 100 test data for 20 random draws. The error bar represents the $95\%$ confidence interval (CI). As expected, as the depth increases, the performance of NTKs drop, eventually their $95\%$ CI becomes a singleton, yet the performance of DEQs stabilize. Also note with larger $\sigma_W^2$, the freezing of NTK takes more depths to happen.

**Result.** On MNIST data, we test the performance of DEQ-NTK with $\sigma_W^2 = 0.25, \sigma_U^2 = 0.25, \sigma_b^2 = 0.5$ and achieve $98.6\%$ test accuracy. The results are listed in table 1.

On CIFAR-10, we trained DEQ-NTK with three different sets of random initializations. These initializations are not fine-tuned, yet we can still see they are comparable, or even superior, to the finite-depth NTK with carefully chosen regularization. For CDEQ-NTK, we compute the kernel matrix on 2000 training data and tested on 10000 samples. See the results in table 2.

**MNIST**

| Method | Model size | Acc. |
|---|---|---|
| DEQ-NTK | | **98.6%** |
| Neural ODE | 84K | 98.2% |
| MON DEQ | 84K | 98.2% |

Table 1: Performance of DEQ-NTK on MNIST dataset, compared to neural ODE (Chen et al., 2018b) and monotone operator DEQ, see these results from Winston & Kolter (2020).

We should emphasize that the calculation of NTK requires a huge amount of computing resource, even for shallow networks. On the other hand, our method provides an efficient way to compute a special NTK with infinite depth. Typically, training a DEQ-NTK on all CIFAR-10 data takes around 400 CPU hour, and the training hour of CDEQ-NTK halves that of its finite-depth CNTK counterpart, as we only need to calculate $\Sigma^{(h)}$, whereas the actual CNTK needs to calculate both $\Sigma^{(h)}$ and $\Theta^{(h)}$.

**CIFAR-10**

| Method | Parameters | Acc. |
|---|---|---|
| DEQ-NTK | $\sigma_W^2 = 0.25, \sigma_U^2 = 0.25, \sigma_b^2 = 0.5$ | 59.08% |
| DEQ-NTK | $\sigma_W^2 = 0.6, \sigma_U^2 = 0.4, \sigma_b^2 = 0$ | **59.77%** |
| DEQ-NTK | $\sigma_W^2 = 0.8, \sigma_U^2 = 0.2, \sigma_b^2 = 0$ | 59.43% |
| NTK with ZCA regularization | $\sigma_W^2 = 2, \sigma_b^2 = 0.01$ | 59.7% |
| DEQ-CNTK with 2000 training data | $\sigma_W^2 = 0.65, \sigma_U^2 = 0.35$ | 37.49% |

Table 2: Performance of DEQ-NTK and CDEQ-NTK on CIFAR-10 dataset, see Lee et al. (2020). for NTK with ZCA regularization.

With a smaller dataset with 1000 training data and 100 test data from CIFAR-10, we evaluate the performance of NTK and the finite depth iteration of DEQ-NTK, as depth increases. See fig. 1 As proven in Jacot et al. (2019), the NTK will always "freeze" in our setting. Therefore the NTK starts to become linearly independent as the depth increases, and its kernel regression does not have a unique solution. To circumvent this issue, we add a regularization term $r \propto \frac{\epsilon \Theta(x,x)}{n}$, where $n$ is the size of the training data. Such regularization is known to guarantee uniform stability (Bousquet & Elisseeff, 2002), and it still interpolates data in the classification sense (training accuracy is $100\%$).

## 6 CONCLUSION

We derive NTKs for both fully-connected DEQs and convolution DEQs, and show that they can be computed more efficiently than finite depth NTK and CNTK, especially when the depth is deep. Moreover, the performance of DEQ-NTK and CDEQ-NTK is comparable to their finite depth NTK counterparts. Our analysis shows that one can avoid the freeze and chaos phenomenon in infinitely deep NTKs by using input injection. One interesting question remained open is to further understand the role of $\sigma_W^2, \sigma_U^2, \sigma_b^2$ in the fixed point computation, and how they affect generalizations of DEQ-NTKs.

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

## A    DETAILS OF SECTION 3

In this section, we give the detailed derivation of DEQ-NTK. There are two terms that are different from NTK: $\Sigma^{(h)}(x, y)$ and the extra $\mathbb{E}_\theta \left[ \left\langle \frac{\partial f(\theta, x)}{\partial U}, \frac{\partial f(\theta, y)}{\partial U} \right\rangle \right]$ in the kernel.

Let us restate the FCNN-IJ here:

Let $d$ be the input dimension, $x, y \in \mathbb{R}^d$ be a pair of inputs, $N_h$ be the width of the $h^{th}$ hidden layers. Let $N_0 = d$ and $N_{L+1} = 1$. Define the FCNN-IJ with $L$ layers as follows:

$$f_\theta^{(h)}(x) = \sqrt{\frac{\sigma_W^2}{N_h}} W^{(h)} g^{(h-1)}(x) + \sqrt{\frac{\sigma_U^2}{N_h}} U^{(h)} x + \sqrt{\frac{\sigma_b^2}{N_h}} b^{(h)}, \ h \in [L]$$

$$g^{(L)}(x) = \sigma(f_\theta^{(L)}(x))$$

where $W^{(h)} \in \mathbb{R}^{N_h \times N_{h-1}}$, $U^{(h)} \in \mathbb{R}^{N_h \times d}$ are the internal weights, and $b^{(h)} \in \mathbb{R}^{N_h}$ are the bias terms. These parameters are chosen using the NTK initialization. Let us pick $\sigma_W, \sigma_U, \sigma_b \in \mathbb{R}$ arbitrarily in this section.

*Proof of theorem 1.* First we note that

$$\mathbb{E}\left[ \left[ f^{(h+1)}(x) \right]_i \cdot \left[ f^{(h+1)}(y) \right]_i \mid f^{(h)} \right]$$

$$= \frac{\sigma_W^2}{N} \sum_{j=1}^N \sigma\left( \left[ f^{(h)}(x) \right]_j \right) \sigma\left( \left[ f^{(h)}(y) \right]_j \right) + \frac{\sigma_U^2}{N} \sum_{j=1}^N x^\top y + \sigma_b^2$$

$$\to \Sigma^{(h+1)}(x, y) \ a.s$$

where the first line is by expansion the original expression and using the fact that $W, U, b$ are all independent. The last line is from the strong law of large numbers. This shows how the covariance changes as depth increases with input injection.

Recall the splitting:

$$\Theta^{(L)}(x, y) = \mathbb{E}_\theta \left[ \left\langle \frac{\partial f(\theta, x)}{\partial \theta}, \frac{\partial f(\theta, y)}{\partial \theta} \right\rangle \right]$$

$$= \underbrace{\mathbb{E}_\theta \left[ \left\langle \frac{\partial f(\theta, x)}{\partial W}, \frac{\partial f(\theta, y)}{\partial W} \right\rangle \right]}_{\textcircled{1}} + \underbrace{\mathbb{E}_\theta \left[ \left\langle \frac{\partial f(\theta, x)}{\partial U}, \frac{\partial f(\theta, y)}{\partial U} \right\rangle \right]}_{\textcircled{2}} + \underbrace{\mathbb{E}_\theta \left[ \left\langle \frac{\partial f(\theta, x)}{\partial b}, \frac{\partial f(\theta, y)}{\partial b} \right\rangle \right]}_{\textcircled{3}}.$$

The following equation has been proven in many places:

$$\textcircled{1} = \sum_{h=1}^{L+1} \left( \sigma_W^2 \underset{(u,v) \sim \mathcal{N}(0, \Lambda^{(h)})}{\mathbb{E}} [\sigma(u)\sigma(v)] \cdot \prod_{h'=h}^{L+1} \dot{\Sigma}^{(h')}(x, y) \right), \quad \textcircled{3} = \sum_{h=1}^{L+1} \left( \sigma_b^2 \cdot \prod_{h'=h}^{L+1} \dot{\Sigma}^{(h')}(x, y) \right)$$

For instance, see Arora et al. (2019). So we only need to deal with the second term $\mathbb{E}_\theta \left[ \left\langle \frac{\partial f(\theta, x)}{\partial U}, \frac{\partial f(\theta, y)}{\partial U} \right\rangle \right]$. Write $f = f_\theta(x)$ and $\tilde{f} = f_\theta(y)$, by chain rule, we have

$$\left\langle \frac{\partial f}{\partial U^{(h)}}, \frac{\partial \tilde{f}}{\partial U^{(h)}} \right\rangle$$

$$= \left\langle \frac{\partial f}{\partial f^{(h)}} \frac{\partial f^{(h)})}{\partial U^{(h)}}, \frac{\partial \tilde{f}}{\partial \tilde{f}^{(h)}} \frac{\partial \tilde{f}^{(h)})}{\partial U^{(h)}} \right\rangle$$

$$= \left\langle \frac{\partial f^{(h)}}{\partial U^{(h)}}, \frac{\partial \tilde{f}^{(h)}}{\partial U^{(h)}} \right\rangle \cdot \left\langle \frac{\partial f}{\partial f^{(h)}}, \frac{\partial \tilde{f}}{\partial \tilde{f}^{(h)}} \right\rangle$$

$$\rightarrow \sigma_U^2 x^\top y \cdot \prod_{h'=h}^{L+1} \dot{\Sigma}^{(h')}(x, y)$$

where the last line uses the existing conclusion that $\left\langle \frac{\partial f}{\partial f^{(h)}}, \frac{\partial \tilde{f}}{\partial \tilde{f}^{(h)}} \right\rangle \rightarrow \prod_{h'=h}^{L+1} \dot{\Sigma}^{(h')}(x, y)$, this convergence almost surely holds when $N \rightarrow \infty$ by law of large numbers.

Finally, summing $\left\langle \frac{\partial f}{\partial U^{(h)}}, \frac{\partial \tilde{f}}{\partial U^{(h)}} \right\rangle$ over $h \in [L]$ we conclude the assertion. $\square$

We now proceed to explain more about the fixed point convergence in theorem 1. Let us first show the limit converges.

**Lemma 1.** *Use the same notations and settings in theorem 1 and theorem 2. $\Theta^{(L)}(x, y)$ in eq. (6) converges absolutely if $\sigma_W^2 < 1$.*

*Proof.* Since we pick $x, y \in \mathbb{S}^{d-1}$, and by DEQ-NTK initialization, we always have $\Sigma^{(h)}(x, y) < 1$ for $x \neq y$. Let $\rho = \Sigma^{(h)}(x, y)$, by eq. (5) and eq. (11), if $\sigma_W^2 < 1$, then there exists $c$ such that $\dot{\Sigma}^{(h)}(x, y) < c < 1$ for all finite number of $x \neq y$ on $\mathbb{S}^{d-1}$, and large enough $h$. This is because $\lim_{h \rightarrow \infty} \dot{\Sigma}^{(h)}(x, y) = \dot{\Sigma}^*(x, y) < \dot{\Sigma}^*(x, x) < 1$.

Use comparison test,

$$\lim_{L \rightarrow \infty} \sum_{h=1}^{L+1} \left| \left( \Sigma^{(h-1)}(x, y) \right) \cdot \prod_{h'=h}^{L+1} \dot{\Sigma}^{(h')}(x, y) \right| < 1 + \lim_{L \rightarrow \infty} \sum_{h=1}^{L+1} c^{L+1-h}.$$

Since $c < 1$, the geometric sum converges absolutely, hence $\Theta^*(x, y)$ converges absolutely if $\sigma_W^2 < 1$, and the limit exists. $\square$

# B  DETAILS OF SECTION 4

We first explain the choice of $S$ in the CDEQ-NTK initialization. In the original CNTK paper (Arora et al., 2019), the normalization is simply $1/q^2$. However, due to the zero-padding, $1/q^2$ does not normalize all $\left[\Sigma^{(h)}(x,x)\right]_{ij,i'j'}$ as expected: only the variances that are away from the corners are normalized to 1, but the ones near the corner are not. $[S]_{ij}$ is simply the number of non-zero entries in $\left[\tilde{X}\right]_{\mathcal{D}_{ij,ij}}$.

Now we give the proof to our main theorem.

*Proof of theorem 3.* Similar to the proof of theorem 1, we can split the CDEQ-NTK in two terms:

$$\Theta^{(L)}(x,y) = \mathbb{E}_\theta\left[\left\langle \frac{\partial f(\theta,x)}{\partial \theta}, \frac{\partial f(\theta,y)}{\partial \theta}\right\rangle\right]$$

$$= \underbrace{\mathbb{E}_\theta\left[\left\langle \frac{\partial f(\theta,x)}{\partial W}, \frac{\partial f(\theta,y)}{\partial W}\right\rangle\right]}_{\text{\textcircled{1}}} + \underbrace{\mathbb{E}_\theta\left[\left\langle \frac{\partial f(\theta,x)}{\partial U}, \frac{\partial f(\theta,y)}{\partial U}\right\rangle\right]}_{\text{\textcircled{2}}}.$$

Omit the input symbols $x, y$, let

$$\left[\widehat{K}^{(h)}\right]_{ij,i'j'} = \frac{\sigma_W^2}{[S]_{ij}\cdot[S]_{i'j'}} \mathop{\mathbb{E}}_{(u,v)\sim\mathcal{N}(0,\Lambda^{(h)}_{ij,i'j'})}[\sigma(u)\sigma(v)].$$

As shown in Arora et al. (2019), we have

$$\left\langle \frac{\partial f_\theta(x)}{\partial W^{(h)}}, \frac{\partial f_\theta(,y)}{\partial W^{(h)}}\right\rangle \to \mathrm{Tr}\left(\dot{K}^{(L)}\odot\mathcal{L}\left(\dot{K}^{(L-1)}\odot\mathcal{L}\left(\cdots\dot{K}^{(h)}\odot\mathcal{L}\left(\widehat{K}^{h-1}\right)\cdots\right)\right)\right)$$

Write $f = f_\theta(x)$ and $\tilde{f} = f_\theta(y)$. Following the same step, by chain rule, we have

$$\left\langle \frac{\partial f}{\partial U^{(h)}}, \frac{\partial \tilde{f}}{\partial U^{(h)}}\right\rangle \to \mathrm{Tr}\left(\dot{K}^{(L)}\odot\mathcal{L}\left(\dot{K}^{(L-1)}\odot\mathcal{L}\left(\cdots\dot{K}^{(h)}\odot\mathcal{L}\left(K^{(0)}\right)\cdots\right)\right)\right)$$

Rewrite the above two equations in recursive form, we can calculate the $L$-depth iteration of CDEQ-NTK by:

- For the first layer $\Theta^{(0)}(x,y) = \Sigma^{(0)}(x,y)$.

- For $h = 1, \ldots, L-1$, let

$$\left[\Theta^{(h)}(x,y)\right]_{ij,i'j'} = \mathrm{Tr}\left(\left[\dot{K}^{(h)}(x,y)\odot\Theta^{(h-1)}(x,y) + K^{(h)}(x,y)\right]_{\mathcal{D}_{ij,i'j'}}\right) \quad (22)$$

- For $h = L$, let

$$\Theta^{(L)}(x,y) = \dot{K}^{(L)}(x,y)\odot\Theta^{(L-1)}(x,y) + K^{(h)}(x,y) \quad (23)$$

- The final kernel value is $\mathrm{Tr}(\Theta^{(L)}(x,y))$.

Using eq. (22) and eq. (23), we can find the following recursive relation:

$$\Theta^{(L+1)}(x,y) = \dot{K}^{(L+1)}(x,y)\odot\mathcal{L}\left(\Theta^{(L)}(x,y)\right) + K^{(h+1)}(x,y) \quad (24)$$

At this point, we need to show that $K^*(x,y) \triangleq \lim_{L\to\infty} K^{(L)}(x,y)$ and $\dot{K}^*(x,y) \triangleq \lim_{L\to\infty} \dot{K}^{(L)}(x,y)$ exist. Let us first agree that for all $h \in [L]$, $(ij, i'j') \in [P] \times [Q] \times [P] \times [Q]$, the diagonal entries of $\Lambda_{ij,i'j'}^{(h)}$ are all ones. Indeed, these diagonal entries are 1's at $h = 0$ by initialization. Note that iterating eqs. (17) to (20) to solve for $[\Sigma^{(h)}(x,y)]_{ij,i'j'}$ is equivalent to iterating $f : \mathbb{R}^{P\times Q\times P\times Q} \to \mathbb{R}^{P\times Q\times P\times Q}$:

$$P^{(h+1)} = f(P^{(h)}) \triangleq \mathcal{L}\left(\frac{1}{[S]_{ij}[S]_{i'j'}} R_\sigma(P^{(h)})\right), P^{(0)} = K^{(0)} \tag{25}$$

where

$$R_\sigma(P_{ij,i'j'}^{(h)}) \triangleq \sigma_W^2 \left(\frac{\sqrt{1 - \left(P_{ij,i'j'}^{(h)}\right)^2} + \left(\pi - \cos^{-1}\left(P_{ij,i'j'}^{(h)}\right)\right) P_{ij,i'j'}^{(h)}}{\pi}\right) + \sigma_U^2 K_{ij,i'j'}^{(0)} \tag{26}$$

is applied to $P^{(h)}$ entrywise.

Due to CDEQ-NTK initialization, if $P_{ij,ij}^{(0)} = 1$ for $i \in [P], j \in [Q]$, then $P_{ij,ij}^{(h)} = 1$ for all iterations $h$. This is true by the definition of $S$.

Now if we can show $f$ is a contraction, then $\Sigma^*(x,y) \triangleq \lim_{h\to\infty} \Sigma^{(h)}(x,y)$ exists, hence $K^*$ and $\dot{K}^*$ also exist. We should keep the readers aware that $f : \mathbb{R}^{P\times Q\times P\times Q} \to \mathbb{R}^{P\times Q\times P\times Q}$, so we should be careful with the metric spaces. We want every entry of $\Sigma^{(h)}(x,y)$ to converge, since this tensor has finitely many entries, this is equivalent to say its $\ell^\infty$ norm (imagine flattening this tensor into a vector) converges. So we can equip the domain an co-domain of $f$ with $\ell^\infty$ norm (though these are finite-dimensional spaces so we can really equip them with any norm, but picking $\ell^\infty$ norm makes the proof easy).

Now we have $f = \mathcal{L} \circ \frac{1}{[S]_{ij}[S]_{i'j'}} R_\sigma : \ell^\infty \to \ell^\infty$. If we flatten the four-way tensor $P^{(h)}$ into a vector, then $\mathcal{L}$ can be represented by a $(P \times Q \times P \times Q) \times (P \times Q \times P \times Q)$ dimensional matrix, whose $(kl, k'l')$-th entry in the $(ij, i'j')$-th row is 1 if $(kl, k'l') \in \mathcal{D}_{ij,i'j'}$, and 0 otherwise. In other words, the $\ell^1$ norm of the $(ij, i'j')$-th row represents the number of non-zero entries in $\mathcal{D}_{ij,i'j'}$, but by the CDEQ-NTK initialization, the row $\ell^1$ norm divided by $[S]_{ij} \cdot [S]_{i'j'}$ is at most 1! Using the fact that $\|\mathcal{L}\|_{\ell^\infty \to \ell^\infty}$ is the maximum $\ell^1$ norm of the row, and the fact $R_\sigma$ is a contraction (proven in theorem 2), we conclude that $f$ is indeed a contraction.

With the same spirit, we can also show that eq. (23) is a contraction if $\sigma_W^2 < 1$, hence eq. (21) is indeed the unique fixed point. This finishes the proof. □

### B.1 COMPUTATION OF CDEQ-NTK

One may wish to directly compute a fixed point (or more precisely, a fixed tensor) of $\Theta^{(L)} \in \mathbb{R}^{P\times Q\times P\times Q}$ like eq. (8). However, due to the linear operator $\mathcal{L}$ (which is just the ensemble of the trace operator in eq. (20)), the entries depend on each other. Hence the system involves a $(P \times Q \times P \times Q) \times (P \times Q \times P \times Q)$-dimensional matrix that represents $\mathcal{L}$. Even if we exploit the fact that only entries on the same "diagonal" depend on each other, $\mathcal{L}$ is at least $P \times Q \times P \times Q$, which is $32^4$ for CIFAR-10 data.

Moreover, this system is nonlinear. Therefore we cannot compute the fixed point $\Sigma^*$ by root-finding efficiently. Instead, we approximate it using finite depth iterations, and we observe that in experiments they typically converge to $10^{-6}$ accuracy in $\ell^\infty$ within 15 iterations.

