# OpenReview forum: "On the Neural Tangent Kernel of Equilibrium Models"
_ICLR.cc/2021/Conference — Reject_

### Official Review · AnonReviewer3 · 2020-10-16

**Rating:** 4
**Confidence:** 3

**Review:**

This paper studies the neural tangent kernel (NTK) of fully-connected neural networks with input injection (defined in the first set of display in Section 3.1), and the infinite depth limit of the NTK. The calculations are further carried out for the convolution neural networks with input injection (defined at the beginning of page 6). Those kernels are empirically evaluated on MNIST and CIFAR-10 datasets, and are compared with the usual NTKs without input injection.

The theorems derived in this paper are incremental given existing studies on NTKs. The calculations are very similar to existing ones except the network structures considered in the current paper are slightly different. The infinite-depth limit of the kernels now indeed depends on the input, but the result is not surprising as the input is injected in each layer.

This paper also lacks a proper introduction to many concepts. For example, it is hard to understand what does DEQ-NTK really means in the introduction. Also the term NTK at first refers to the general concept in (1), but later seems to specifically refer to the NTK of fully-connected neural networks without input injection. Section 2 presents some background, but it gives many pieces of related works without a clear structure. For instance, I don't see how many concepts like "weakly-trained", "fully-trained", "edge-of-chaos" are relevant to the current paper.

In the experiments, the choices of parameters seem to be very arbitrary. The authors do not provide systemic guidance on how they tune those parameters, while the performance improvement is minor. Since one major motivation for studying NTKs is the relation to the actual neural networks, some proper comparisons or comments should be included. The experiment section in the current form is not very convincing.

Given the above concerns, I don't think this paper is suitable for publication in ICLR.

---

> ### Author Response · Authors · 2020-11-16
> **Thanks for your review**
>
>  Thanks for your review and suggestions, we appreciate your feedback.
>
> The proofs are indeed incremental, but they are not the focus of this paper. Instead, the main focus is the concept of what happens to DEQ in the infinite width limit, and how can we calculate the limits using root-finding. The choice of parameters is indeed arbitrary in our experiment section. On a high level, one can imagine how $\sigma_W^2$ and $\sigma_U^2$ give tradeoff between the previous layer and the input injection, but in experiments, we notice they don't make a huge difference. One should note that in most NTK papers, these parameters are not tuned systematically, e.g. see Finite Versus Infinite Neural Networks: an Empirical Study (\url{https://proceedings.neurips.cc/paper/2020/file/ad086f59924fffe0773f8d0ca22ea712-Paper.pdf}).
>
> We will also try to make our concepts and intro section more clear.

---

### Official Review · AnonReviewer2 · 2020-10-27
**Interesting mix of techniques for a specific kind of models**

**Rating:** 6
**Confidence:** 4

**Review:**


This paper studies the double infinite-width + infinite-depth limit of fully connected and convolutional neural nets from an NTK angle, when input injections enter the picture. The techniques mix NTK techniques with Deep Equilibrium (DEQ) model techniques to directly compute the infinite-depth limit of the infinite-width limit of such neural nets. They show that there is no freeze/chaos transition for such networks (unlike the case without input injections). The writing is reasonably clear, although the size of the formulas is not very pleasant.
The experimental part is not very detailed, and it is not clear what is the take-home message from it.
Pros: This is quite interesting, the technique is nice.
Cons: the scope is somehow limited to a class of models which is not very much used; (as the authors say), the role of the normalizations factors that appear is not super clear; no very surprising phenomena, somehow.
Overall, I think that these results are mathematically interesting and could lead in principle to practically useful insights, although this is not realized at this point. There could be some presentation effort in terms of the sizes of the formulae: what do we really need to know from them?

---

> ### Author Response · Authors · 2020-11-16
> **Thanks for your review**
>
> Thanks for your review and suggestions, we appreciate your feedback.
>
> "models are not widely used": this is indeed true. DEQs are not as widely used as other structures like CNN/ResNet etc. However, one should note that they achieve near SOTA performance with only constant memory, so we expect they will become more popular.
>
> The takeaways from the experiment section are the following: 1) we indeed see that as the depth increases, NTK of vanilla network freezes, but DEQ-NTK stabilizes, as our theorems show. 2) Performance of DEQ-NTK is comparable to that of NTK of finite depth FCNN without input injection, this is showing that the DEQ-NTK is not meaningless. 3) The derivation is general enough and can be applied to other structures like CNN.

---

### Official Review · AnonReviewer4 · 2020-10-28
**Recommendation to reject on "On the Neural Tangent Kernel of Equilibrium Models"**

**Rating:** 3
**Confidence:** 4

**Review:**

The paper shows the deep equilibrium model has non-degenerate neural tangent kernel in the infinite depth setting. The neural tangent kernel can be computed by a similar root-finding problem as that in the deep equilibrium problem itself. Some experiments have been performed to compare the performance of deep equilibrium neural tangent kernel with that of finite depth neural tangent kernel.

Overall I vote for rejecting. My concerns are as follows:

The paper lacks related literature. First, the motivation of considering deep equilibrium models is unclear to me. The authors should provide some further literature review. The advantage of using such a model in practice should be explained. Second, related proof techniques in the existing literature needs to be discussed.

The result is expectable and the proof techniques are not novel. The main theorem (Theorem 1) is the simple extension of the existing results on neural tangent kernel. The following theorem (Theorem 2) is the consequence of the main theorem under some specified initialization.

The theorems in the paper are lack of explanation. More discussion is needed to explain and extend the results in the paper.

The experiment part is not well-organized. More description is needed to improve the results.

The paper has some grammar mistakes and misuse of words. The paper needs to be revised carefully. To name a few:
Abstract: DEQ model....DEQ models have...
Section 3, 1st paragraph: we simplify fully-connected DEQs as DEQs.
Section 3, 1st paragraph: In section 3.1, we show the NTK of the approximated DEQ using finite depth iteration...

---

> ### Author Response · Authors · 2020-11-16
> **Thanks for your review**
>
> Thanks for your review and suggestions, we appreciate your feedback.
>
> Why use DEQ: as mentioned in the introduction, they achieve SOTA performance, while requiring only constant memory (unlike vanilla NN, which stores gradients of each layer).
>
> Lack of novelty in proof: the proof is not very hard, but we want to emphasize that this is not a theory paper. Instead, we try to provide a concept of what will happen when the width of DEQ goes to infinity.
>
> We will try to make the related work section more clear. Thanks for pointing out other mistakes, we will modify them.

---

### Official Review · AnonReviewer1 · 2020-10-28
**This paper derives NTK for DEQ models and provides several comparisons with NTK and CNTK. The design of the experiments has serious flaws, which makes it hard to estimate the value of the contribution.**

**Rating:** 4
**Confidence:** 4

**Review:**

This paper considers the problem of deriving NTK for DEQ models and shows that DEQ models have non-degenerate NTKs even in the infinite depth limit. It also provides several experimental comparisons on the performance of the obtained DEQ-NTK and CDEQ-NTK with NTK and CNTK, respectively.

Strengths:

1) The derivation of NTK for DEQ models seems to be reasonable and correct, and the resulting DEQ models have non-degenerate NTKs.

2) By using the rooting-finding ability of the DEQ models, these derived NTK kernels can be computed for both fully-connected and convolutional variants.

Weaknesses:

1) The experiment design has some serious flaws. For example, the experiments include comparisons only with NTK and CNTK. It would be natural to also include comparisons with DEQ models, as DEQ-NTK (or CDEQ-NTK) can be viewed as augmented model of DEQ (or CDEQ). Thus, it would be interesting to see whether the new models can indeed achieve better performance. Otherwise, it would be no point to use NTK on DEQs.

2) Particularly, the paper misses comparisons with several important DEQs models, like MON DEQ and Single stream DEQ,  on CIFAR-10 and MNIST.

          https://arxiv.org/abs/2006.08591​ Monotone operator equilibrium networks
          https://arxiv.org/abs/2006.08656​ Multiscale Deep Equilibrium Models
          https://arxiv.org/abs/1909.01377​ Deep equilibrium models

   These DEQ models achieve much better experimental results than the reported ones in this paper. For example,
        CIFAR-10:
           MON DEQ: single conv 74.1%
          Single stream DEQ: around 82.2%
     MNIST:
         MON DEQ: single conv 99.2%

  Missing these comparisons intentionally or unintentionally significantly weakens the paper.

3) This paper also misses the comparison between CNTK  and CDEQ-NTK on  CIFAR-10.

     CDEQ-NTK with 2000 training set result: 37.49% --- reported in this paper
     CNTK (vanilla) with 2000 training set: 40.94% (Depth 3), 42.54%(Depth 4), 43.43%(Depth 6), 43.42%(Depth 11), 42.53%(Depth 21).
      https://arxiv.org/abs/1904.11955​ On Exact Computation with an Infinitely Wide Neural Net

It would strengthen the paper if it can show that CDEQ-NTK achieves better performance than CNTK when the number of layers increases. This is also what the paper tries to claim.

4)  The DEQ-NTK in the experiment is derived for FCNN. From the first experiment, one can only see that DEQ-NTK achieves better performance than NTK for FCNN when the depth is large.  Since it makes more sense to apply DEQ to a complicated neural network (like transformer or trellis net) than to an FCNN, it leaves a doubt whether the proposed DEQ-NTK has any  practical importance. Thus, it would be better if the paper conducts experiments for the more complicated networks.

5) It is not surprising to see that when the depth increases, DEQ-NTK remains stable while NTK does not, as this seems to be enabled more by the mechanism of the DEQ model.

6) Related work is not sufficiently discussed. For example, the freezing of NTK.

---

> ### Author Response · Authors · 2020-11-16
> **Thanks for your review**
>
> Thanks for your review and detailed suggestions. We appreciate your feedback.
>
> Comparison with DEQs: we leave these comparisons out on purpose. It is not surprising that for image classfication tasks, neural networks typically outperform their NTK counterparts (and sometimes by a lot). We want to emphasize that this work (along with other NTK papers) is not trying to come up with useful models that achieve better performance, rather it tries to gain insight about DEQs. In this paper, we learn that even in the inifinite-width regime, DEQ-NTKs are not degenerate like vanilla NTKs, and this should be the main takeaway. As a side note, there's no point using NTKs in reality, and people don't use them in practice, but it is indeed an important limiting scenario.
>
> Comparison with CNTK: we want to point out that we are not claiming that increasing depth always gives a superior performance. In fact, in the paper you mentioned (On Exact Computation with an Infinitely Wide Neural Net), the authors observed that for CNTK (both with and without GAP), 11 depths have better performance than 21 depth. Our claim about the CDEQ-NTK is the following: 1) the derivation of DEQ-NTK (i.e. the NTK for fully-connected nn with input injection) is general and can be applied to other structures; and 2), if in scenario people want to compute a very deep NTK yet have computational constraints, our model provides a more practical way (as compared to actually do the forward iterations) to compute a variant NTK.
>
> "Derivation for complicated networks": these are indeed interesting topics, but we argue that the idea of NTK is conceptual rather than practical. Even if one derived NTKs for trellis net, it's likely that trellis net will outperform its NTK counterpart.
>
> "..it's not surprising..": The phenomenon itself may not be surprising, but it's interesting to show how one can use root finding to solve the NTK. We also want to point out that unlike DEQ, which may not guarantee a stable fixed point (unless you use other parameterization like monotone DEQ), the infinite width regime has a much simpler convergence requirement ($\sigma_W^2<1$).

---

### Decision · Program_Chairs · 2021-01-07
**Final Decision**

**Decision:**

Reject

**Comment:**

This paper combines recently emerging NTK theory and kernels with DEQ models. In particular the authors use the root-finding capability of DEQ models to compute the corresponding NTK of DEQ models for fully connected and convolutional variants. The reviewers raised various concerns including lack of experimental details, incremental theoretical results(which the authors agree with but postulate that this is a practical paper), lack of proper literature review, explaining how it applies in practical scenarios and grammatical mistakes. Some of these concerns were addressed during the response period but none of the reviewers were fully satisfied with the author's response. While I think there are interesting ideas in this paper I agree with the reviewers that a substantial revision is required and therefore recommend rejection.